# Virological Markers in Epstein–Barr Virus-Associated Diseases

**DOI:** 10.3390/v15030656

**Published:** 2023-02-28

**Authors:** Julien Lupo, Aurélie Truffot, Julien Andreani, Mohammed Habib, Olivier Epaulard, Patrice Morand, Raphaële Germi

**Affiliations:** 1Institut de Biologie Structurale, Université Grenoble Alpes, UMR 5075 CEA/CNRS/UGA, 71 Avenue des Martyrs, 38000 Grenoble, France; 2Laboratoire de Virologie, CHU Grenoble Alpes, CS 10217, CEDEX 09, 38043 Grenoble, France; 3Service de Maladies Infectieuses, CHU Grenoble Alpes, CS 10217, CEDEX 09, 38043 Grenoble, France

**Keywords:** Epstein–Barr virus, serology, EBV DNA, biomarkers, infectious mononucleosis, lymphoma, nasopharyngeal carcinoma, post-transplant lymphoproliferative disorders, multiple sclerosis

## Abstract

Epstein–Barr virus (EBV) is an oncogenic virus infecting more than 95% of the world’s population. After primary infection—responsible for infectious mononucleosis in young adults—the virus persists lifelong in the infected host, especially in memory B cells. Viral persistence is usually without clinical consequences, although it can lead to EBV-associated cancers such as lymphoma or carcinoma. Recent reports also suggest a link between EBV infection and multiple sclerosis. In the absence of vaccines, research efforts have focused on virological markers applicable in clinical practice for the management of patients with EBV-associated diseases. Nasopharyngeal carcinoma is an EBV-associated malignancy for which serological and molecular markers are widely used in clinical practice. Measuring blood EBV DNA load is additionally, useful for preventing lymphoproliferative disorders in transplant patients, with this marker also being explored in various other EBV-associated lymphomas. New technologies based on next-generation sequencing offer the opportunity to explore other biomarkers such as the EBV DNA methylome, strain diversity, or viral miRNA. Here, we review the clinical utility of different virological markers in EBV-associated diseases. Indeed, evaluating existing or new markers in EBV-associated malignancies or immune-mediated inflammatory diseases triggered by EBV infection continues to be a challenge.

## 1. Introduction

Epstein–Barr virus (EBV), also known as human herpesvirus type 4 (HHV-4), belongs to the *Herpesviridae* family and the *Gammaherpesvirinae* subfamily. EBV is an enveloped double-stranded DNA virus with a diameter of 150 to 200 nm and approximately 180,000 base pairs that code for 80 to 100 proteins [1]. EBV was first discovered in Burkitt lymphoma tumor cells in 1964 [2].

EBV infects more than 95% of adults worldwide. Primary infection is usually asymptomatic, although it can cause infectious mononucleosis (IM), most often in young adults. In the early phase of primary infection, most EBV lytic genes, which encode very early, early, and late proteins, are expressed: this defines the lytic or productive infection in which virions containing a linear DNA genome are produced. EBV uses B-cell activation and differentiation for primary infection and persistence in lymphoid tissue. During the latency phase, the virus genome is a self-replicating episome in the cell nucleus, where it can express a number of latency proteins associated with different transcriptional programs (Table 1). It is now acknowledged that some lytic proteins (i.e., ZEBRA, v-IL10, BHRF1, BALF1) are crucial for the establishment of viral latency [3].

After primary infection, EBV persists lifelong in the memory B lymphocytes of the infected host, with episodic reactivations in saliva without clinical consequences. This results in a balance between viral replication and host immune responses. However, EBV persistence combined with genetic and other factors can lead to the development of cancers or trigger immune-mediated inflammatory diseases such as multiple sclerosis (MS). EBV is associated with epithelial cancers such as nasopharyngeal carcinoma (NPC) and 10% of gastric cancers, in addition to lymphomas, including Burkitt lymphoma (BL), Hodgkin lymphoma (HL), diffuse large B-cell lymphoma (DLBCL), and natural killer (NK)/T-cell lymphoma (NKTCL) as well as post-transplant lymphoproliferative disorders (PTLD) [4]. Indeed, in some infected B cells with genetic alterations, EBV can interfere with cellular homeostasis by promoting immune evasion, cell growth, or genetic instability and inhibiting apoptosis [5]. EBV can exert its oncogenic properties in immunocompetent hosts and immunosuppressed patients in whom the inhibition of anti-EBV immune responses plays a key role in the development of EBV-induced lymphoproliferative disorders [6]. For more details, the readers are referred to constructive reviews that describe the mechanisms of virus-associated oncogenesis [7,8].

Standard markers of EBV involve EBV DNA and anti-EBV antibody detection in blood compartments. Historically, EBV serology contributed to identify causality links between EBV infection and diseases suspected to be attributable to the virus. Currently, in clinical practice, EBV serology is mainly limited to the diagnosis of IM, the determination of EBV status, and NPC screening with the exploration of specific anti-EBV immunoglobulin A (IgA) markers. Further studies are needed to validate the use of EBV antibodies in other clinical contexts. Several researchers investigating EBV DNA in EBV-associated malignancies, including NPC, provide strong evidence for its application in clinical practice. In lymphoma, even if EBV DNA appears to be a promising prognostic marker, some results remain controversial. Its use in clinical practice therefore requires further evaluation.

This paper reviews the use of virologic markers in clinical practice for the main EBV-associated diseases. The contribution of new markers or technologies such as next-generation sequencing for the diagnosis of EBV-associated diseases will also be discussed.

## 2. Infectious Mononucleosis

EBV primary infection is mainly asymptomatic in young children but can cause IM in 25% to 79% of infected adolescents and young adults [9,10]. Indeed, later infection increases the risk of developing IM [11]. The virus is mainly transmitted by saliva [12] but can also be transmitted by hematopoietic stem cell transplantation (HSCT) or solid organ transplantation (SOT).

IM symptoms occur 30 to 50 days after transmission and usually include fever, angina with or without nasopharyngitis, lymphadenopathy, splenomegaly, and severe fatigue for a period of 2 to 6 weeks. Symptoms are mainly due to inadequate anti-EBV CD8 T-cell and NK immune responses [13]. From a biological perspective, the classic mononucleosis syndrome can be associated with moderate hepatic cytolysis.

IM is mostly benign with symptoms resolving within a month. Only 10% to 20% of patients present prominent initial symptoms (e.g., major fatigue, dysphagia, hepatic or hematological disorders), and in less than 1% of cases, life-threatening complications can occur [14,15]. Severe IM may occur in patients with a specific genetic and immunological background [16]. X-linked lymphoproliferative syndrome, also known as Purtilo syndrome or Duncan disease, is fatal in approximately 60% of cases. It is a consequence of mutations in the SH2D1A gene, which encodes the SLAM-associated protein involved in the activation of cytotoxic T lymphocytes [16]. Chronic active EBV infection, which is associated with CD8 and NK/T lymphoproliferations [17,18], has a poor prognosis. The geographical distribution of this disease, which is mainly observed in Asia, suggests a genetic predisposition. Chronic active EBV infection is suspected in the case of persistent IM symptoms and a very high EBV viral load.

Changes in IM epidemiology have been observed in industrialized countries since the 2000s, linked notably to the older age of patients at the time of primary EBV infection, thus leading to an increase in severe and complicated forms [19,20,21,22]. Epidemiological studies have also shown that IM may be associated with a greater risk of developing MS [23] or EBV-positive HL [24]. Nevertheless, it cannot be excluded that IM, MS, and HL share the same genetic and immunological determinants.

The virological diagnosis of IM is based on EBV-specific serology, as its clinical symptoms are unspecific. The mononucleosis syndrome is present in other primary infections such as cytomegalovirus, human immunodeficiency virus (HIV), and toxoplasmosis [25]. The detection of heterophile antibodies by rapid tests can be useful when EBV serology is difficult to interpret. These non-specific IgM antibodies only appear in IM, being associated with immune hyperstimulation and non-specific B lymphocyte activation. Heterophile antibodies do not target EBV antigens but react against red blood cell antigens. They are usually detected along with IM symptoms (or sometimes after a slight delay) and persist for up to 3 months after the onset of IM (Figure 1). The specificity of heterophile antibody tests exceeds 95%, with false positives being exceptional. These tests can therefore be used alone in the case of typical symptoms and/or standard laboratory results. On the contrary, test sensitivity is low (85% and even less than 50% in children under 6 years and in adults over 50 years). As a negative result for heterophile antibodies does not eliminate the diagnosis of IM, testing should be completed with EBV-specific serology [26,27].

EBV-specific serology mainly includes the exploration of three or four markers using automated immunoassays: anti-viral capsid antigen (VCA) or total anti-EBV immunoglobulin M (IgM), anti-VCA, anti-early antigen (EA) or total anti-EBV immunoglobulin G (IgG), and anti-Epstein–Barr nuclear antigen (EBNA)-1 IgG. The combination of these markers can differentiate a primary infection from a past infection. EBV primary infection is characterized by the presence of anti-VCA IgM with or without anti-VCA IgG but always without anti-EBNA-1 IgG. Indeed, anti-EBNA-1 IgG appears 2–3 months after symptom onset. Anti-VCA IgG may be absent at the very start of symptoms. Like heterophile antibodies, anti-VCA IgM persists for 2–3 months after symptom onset, although it may be detected for a longer period. Anti-VCA IgG persists for life (Figure 1 and Table 2). Past infection is defined by the presence of anti-VCA IgG and anti-EBNA-1 IgG without anti-VCA IgM. However, in 3–5% of individuals and 10–20% of immunocompromised patients, anti-EBNA-1 IgG are not detected after primary infection [28,29,30]. This so-called “unbalanced” serological profile is not related to specific symptoms or risks.

Other profiles can be more difficult to interpret, thus being labelled as “unusual” or “indeterminate” cases. Here, the clinical context (age, symptoms) is helpful, and serological follow-up is essential. Further diagnostic approaches based on serology (immunofluorescence or immunoblot) or polymerase chain reaction (PCR) can elucidate challenging profiles [28,31,32]. EBV DNA in serum or plasma is detected in the first 7 days after the onset of IM but never after the 15th day. If positive, it can reinforce the diagnosis of a recent primary infection. EBV PCR in saliva can also be helpful, as it always exceeds 1 million copies/mL during the first few months after IM [12]. However, high viral loads in saliva can also be detected in EBV reactivations in immunocompromised patients. EBV serology or PCR in cerebrospinal fluid can be of interest in cases of neurological complications due to IM. In transplant recipients, especially EBV seronegative patients receiving EBV seropositive grafts, the detection of antibodies can be delayed after primary infection, and in some cases, the detection of EBV DNA in whole blood can confirm the diagnosis instead of serology.

**Table 2 viruses-15-00656-t002:** Main Epstein–Barr virus serological profiles.

Anti-VCA IgG	Anti-VCA IgM	Anti-EBNA IgG	Interpretation
−	−	−	Seronegative individual
−/+	+	−	Primary infection
+	−	+	Past infection
+	−	−	Past infection (adults) or primary-infection (children) *
+	+	+	Past infection or end of primary infection **
−	−	+	Indeterminate

Abbreviations: VCA: viral capsid antigen; EBNA: Epstein–Barr nuclear antigen.* See references [28,29,30] ** See references [28,32].

A recent study suggested that rapid antibody responses to EBV are correlated with a reduced severity of primary infection. Thus, the time to the onset of serological markers may have a prognostic value [33].

For the past few years, particular attention has been paid to the exploration of neutralizing antibodies in viral infections. It was shown that neutralizing antibodies increase over many months after EBV primary infection [34] and that the EBV viral load in blood and EBV gp350 diversity are important elements for generating a strong neutralizing antibody response [35].

## 3. EBV-Associated Malignancies

EBV is associated with several lymphoid malignancies such as BL, HL, non-Hodgkin lymphoma (NHL), and PTLD as well as epithelial cancers such as NPC and gastric carcinoma. In these pathologies, EBV is an important factor combined with genetic, environmental, immune, or infectious factors that contribute to the development of cancer. Depending on the type of cancer, the level of the association with EBV differs (Table 1). For example, EBV is associated with NPC, endemic BL, T/NK lymphoma, and HIV-associated primary central nervous system lymphoma in 100% of cases contrary to HL (30%), NHL (5–15%), and gastric carcinoma (10%). The diagnosis of EBV-associated cancer is based on pathological results and the detection of RNA (EBER non-coding RNA) and oncogenic viral proteins (LMP-1 and EBNA-1) in the tumor. Blood EBV DNA may be correlated with EBV in situ and have a diagnostic and/or prognostic value in several carcinomas or lymphomas. Specific EBV serological patterns can also be found in some cancers.

### 3.1. Carcinomas

#### 3.1.1. Nasopharyngeal Carcinoma

EBV is associated with undifferentiated NPC, which is endemic to Southern China and Southeast Asia. NPC is responsible for a global burden of around 80,000 new cancer cases annually. Irrespective of its geographical distribution and incidence rate, all cases of undifferentiated NPC worldwide are EBV-associated (i.e., presence of EBV genome in every malignant cell). Tsao et al. postulated that persistent EBV infection in a genetically aberrant epithelial cell can induce tumorigenic transformation and clonal expansion of infected cells [36]. Other co-factors such as long-term exposure to environmental carcinogens contribute to the pathogenesis of NPC, inducing a number of somatic genetic alterations in epithelial cells [36].

Without vaccines to prevent EBV infection and associated diseases, efforts should be made to identify viral biomarkers and to detect individuals at risk of developing NPC or in an early stage of the disease when treatment is most effective [37].

In NPC patients, circulating EBV DNA molecules in blood exist as naked DNA fragments [38]. EBV DNA released from cancer cells into plasma is a well-recognized prognostic biomarker of NPC with a potential clinical value for screening, prognosis, and surveillance of recurrence [39]. In 2017, a large-scale study screened circulating EBV DNA in blood in 20,178 adults aged between 40 and 62 years in Hong Kong and validated the use of plasma EBV DNA analysis for screening NPC [40]. The results showed an NPC detection rate of 168.5/100,000, with a positive and negative predictive value of 11% and 99.99%, respectively, as well as high sensitivity (97.1%) and specificity (98.6%). For the majority of patients, EBV DNA was detected at an early stage of the disease (stage I or II), and earlier detection was associated with longer survival (97% with a 3-year progression-free survival rate). Given that approximately 5% of healthy individuals harbor EBV DNA in their plasma [41], EBV DNA follow-up was shown to reduce false positives [40]. In NPC patients, EBV DNA was continuously released from cancer cells into the blood, whereas non-NPC subjects tended to have transiently positive results [42]. A recent study demonstrated that NPC patients have plasma EBV DNA of different sizes compared to non-NPC subjects (size-based analysis). Plasma EBV DNA from NPC patients exhibited a typical nucleosome size profile with a peak of around 150–160 bp, whereas EBV DNA from non-NPC subjects had smaller fragment lengths. This approach demonstrated a superior performance for NPC screening [43].

Several studies showed the effectiveness of plasma EBV DNA measured before treatment as a prognostic marker in NPC patients. Patients with advanced stage NPC had higher plasma EBV DNA than those with early stage disease [44,45,46,47]. Plasma EBV DNA also demonstrated a positive linear relationship with the total tumor volume [38,48]. Some studies revealed that both pretreatment viral load and cancer stage are independent prognostic factors for overall survival in multivariate analyses [44,49,50]. It was therefore suggested to incorporate pretreatment plasma EBV DNA load into the current staging system of NPC [51].

Radiotherapy is considered as the mainstay treatment for non-metastatic disease. After initiating NPC treatment, patients with detectable EBV DNA mid-treatment are an at-risk group with an unfavorable treatment response. This highlights the prognostic value of the in vivo dynamics of plasma EBV DNA. It therefore seems that patients with faster plasma EBV DNA clearance—and thus undetectable EBV DNA mid-treatment—have more radiosensitive tumors [52,53].

Plasma EBV DNA measured after treatment reflects the residual tumor load after the completion of treatment [54,55]. In a study involving 170 NPC patients at different disease stages, post-treatment EBV DNA load (at 1 month after the completion of treatment) allowed the prediction of recurrence and progression-free survival [54].

Another clinical application of plasma EBV DNA is the surveillance of NPC recurrence. Viral load is expected to be undetectable after curative treatment and tumor eradication. Any increase in the plasma EBV DNA concentrations in the subsequent follow-up period could potentially signify disease relapse. Regular plasma EBV DNA measurement every 3–6 months should be used as an adjunct to endoscopy and imaging in the clinical surveillance protocol of treated NPC patients [56].

Historically, the majority of screening biomarker research has focused on how anti-EBV antibodies can inform us about the risk of NPC. This approach is based on the hypothesis that an increased immune response to EBV is associated with poor viral control and thus disease susceptibility. This is especially true for IgA antibodies produced in the oral cavity where EBV is transmitted and periodically reactivated. An important study conducted in over 9,500 Taiwanese men demonstrated that high anti-VCA IgA levels preceded the development of NPC. Specifically, men initially testing positive for anti-VCA IgA were around 22 times more likely to develop NPC during follow-up [57]. In cohorts of high-risk NPC families, Yu et al. observed elevated levels of both anti-VCA IgA and anti-EBNA-1 IgA before NPC diagnosis. Individuals who were positive for anti-EBNA-1 IgA at baseline experienced higher rates of NPC during follow-up than those testing negative (relative risk = 6.6, 95% confidence interval (CI): 1.5–61) [58]. Another recent study identified the EBNA-1 IgA as a prediagnostic marker of the risk of NPC several years before clinical diagnosis [59].

EBV neutralizing antibodies, and particularly neutralizing antibodies against EBV gp350 (glycoprotein), can be useful to predict the risk of NPC. Neutralizing antibodies that block EBV B-cell infections through gp350 binding are present at significantly higher levels in disease-free control subjects compared to NPC patients (*p* < 0.03) in high-risk NPC families in Taiwan [60]. Family members with both low EBV neutralizing antibodies and elevated anti-EBNA-1 IgA had a seven-fold increased risk of NPC (95% CI: 1.9–28.7). However, in a cohort from the general population in Taiwan, elevated levels of total or IgA-specific gp350 antibody were not protective against the future risk of NPC [61].

In genetically high-risk families, researchers measured IgG and IgA against 199 peptide sequences from 86 EBV proteins in NPC patients and cancer-free control individuals. The risk stratification included anti-VCA IgA and anti-EBNA IgA as well as 12 additional anti-EBV antibodies (including BXLF1, LF2, BZLF1, BRLF1, EAd, BGLF2, BPLF1, BFRF1, and BORF1). This EBV-based risk score improved the prediction of NPC (89%; 95% CI: 82–96%) compared to the current biomarkers (78%; 95% CI: 66–90%; *p* < 0.03) [62,63].

Other studies validated a comprehensive risk prediction model for the accurate stratification of NPC risk using a score that combines EBV and human genetic variants, along with key epidemiological risk factors. This score substantially increased the accuracy of NPC screening when combined with the current standard EBV-serology-based screening [64].

EBV has also been detected in oropharyngeal cancer biopsies and may contribute to the development of EBV-associated oropharyngeal cancer, in combination with other well-established environmental factors such as human papillomavirus infection, tobacco and alcohol [65].

#### 3.1.2. Gastric Carcinoma

Gastric carcinoma (GC) is a major malignancy that affects all populations across the globe. More than 80,000 EBV-positive GCs occur worldwide annually, with the EBV association rate estimated at 10%. EBV-associated GC is distinguished from other GC by a substantial lymphoid infiltrate. In EBV-associated GC, all malignant cells within the tumor carry the same clonal EBV genome. Each tumor developed from a single EBV-infected progenitor cell and was not infected after the malignant state [66].

EBV-positive GC has a better prognosis compared to EBV-negative GC, as it responds remarkably well to immune checkpoint inhibitors. Thus, screening all cases of GC for EBV is highly recommended [67]. As high plasma EBV DNA load in EBV-positive GC patients can predict disease relapse and poor response to chemotherapy, the dynamics of plasma EBV DNA load may be an accessible biomarker to monitor EBV-positive GC [68,69].

Unlike NPC, the link between anti-EBV antibodies and GC risk is unclear. Larger studies with adequate statistical power are needed to evaluate the association between EBV serological markers and GC [70].

### 3.2. Lymphomas

EBV DNA was shown to be present in the blood of patients with other EBV-associated neoplastic disorders, including HL, DLBCL, BL, extranodal NKTL (ENKTL), and PTLD.

#### 3.2.1. EBV-Positive Lymphoma in Immunocompetent Hosts

##### Hodgkin Lymphoma

HL is one of the most common lymphomas in the Western world, with an annual incidence of approximately three new cases per 100,000 people. A recent French study showed an association with EBV in 30% of adult non-HIV patients with classical HL (cHL), defined by the presence of EBV-encoded small RNA (EBER) in Hodgkin Reed–Sternberg tumor cells [71]. The history of IM has been associated with a higher risk of EBV-positive HL [24]. The complex role of EBV in the pathogenesis of cHL has been reviewed elsewhere [72]. In short, pre-apoptotic germinal center B cells may be rescued by EBV, which provides the necessary signals required for cellular growth, survival, and cellular genetic alterations, thus generating a pool of cells that may become progenitors of Hodgkin Reed–Sternberg cells.

In EBV-associated cHL, EBV exists in serum or plasma as episomal or naked EBV DNA derived from predominantly lymphoma cells during the process of apoptosis/necrosis [73]. High EBV DNA loads in blood were previously described in EBV-positive cHLs [74]. Recent data revealed that before treatment, positive EBV DNA load in plasma and whole blood of non-HIV cHL correlated with EBER detection and a pejorative prognostic marker of the disease [75,76,77,78]. In a large American cohort of adult patients with cHL, Kanakry et al. demonstrated that plasma EBV DNA before treatment and at 6-month follow-up was an independent prognostic marker of the outcome of cHL [75]. The same authors observed that the presence of EBV DNA in plasma 8 days after treatment initiation, but not at diagnosis, predicted inferior event-free survival in children and adolescents with cHL [76]. Hohaus et al. reported that plasma EBV DNA load at diagnosis of cHL in 93 Italian patients was associated with a significantly shorter progression-free survival in univariate analysis but not after multivariate Cox proportional hazard regression analysis [79]. Overall, these findings suggest that plasma EBV DNA detection may be a promising marker of prognosis and treatment response in patients with EBV-associated cHL. Nevertheless, the current guidelines do not recommend the use of EBV DNA in the management of these patients with cHL.

Historical evidence supports the link between atypical EBV serological patterns and the risk of EBV associated-cHL [80]. High anti-VCA, anti-EA, and anti-EBNA IgG levels were specifically associated with a high risk of EBV-positive HL but not EBV-negative HL. EBV-associated cHL was also significantly associated with a low anti-EBNA-1/anti-EBNA-2 antibody ratio [81]. A recent study explored the antibody response against the complete EBV proteome in cHL patients, with the results showing different EBV serological patterns between EBV-associated cHL and EBV-negative cHL or control individuals [82]. Antibody response against BdRF1(VCAp40)-IgG and BZLF1(Zta)-IgG were identified as the best serological markers to distinguish EBV-positive from EBV-negative cHL tumors. Liu et al. also showed an elevated antibody response against LMP-1, BHRF1, and BARF1, which are three proteins involved in oncogenesis [82]. Future studies are required to better understand why some individuals cannot control EBV infection and how antibody responses can be used for risk stratification.

##### Burkitt Lymphoma

Among BL variants, endemic BL is associated with EBV in 95–100% of cases. Its geographical distribution in equatorial Africa corresponds to the “lymphoma belt” described historically by Denis Burkitt who, along with Epstein and Barr, contributed to the discovery of the first human oncogenic virus, namely, EBV. The pathogenesis of endemic BL includes genetic factors (c-myc translocation) as well as EBV and Plasmodium falciparum coinfection, which act synergistically in the development of the tumor [83].

Several studies showed high EBV DNA load in circulating cell-associated specimens or plasma from children with endemic BL [84,85]. Median EBV DNA load in plasma was higher in children with BL than in those without lymphoma, although the EBV DNA detection rate did not differ between BL patients and control subjects [86].

Early reports following the discovery of EBV in addition to more recent research demonstrated that Ugandan children with higher levels of antibody against the VCA antigen were more likely to develop BL [87,88]. Recently, Coghill et al. conducted a serological study in Ghanese children using a multiplex technology targeting peptide sequences from 86 EBV proteins [89]. The authors showed that 33 anti-EBV IgG responses were elevated in BL children compared to control individuals. High IgG levels were associated with EBV proteins involved in viral replication and antiapoptotic signaling such as BMRF1 (early antigen), BBLF1 (tegument protein), BHRF1 (Bcl-2 homolog), BZLF1 (ZEBRA), BILF2 (glycoprotein), BLRF2 (VCAp23), BDLF4, and BFRF3 (VCAp18). This approach could provide new insights into the pathogenesis of BL.

##### Diffuse Large B-Cell Lymphoma

DLBCL represents 30% of all cases of NHL. The prevalence of EBV-positive DBLCL is only 5% in Western countries and 10–15% in Asia and South America. EBV-positive DLBCL was included in the 2016 World Health Organization (WHO) classification of lymphoid neoplasms. In the era of chemoimmunotherapy, EBV-positive DLBCL is an aggressive B-cell lymphoma with a worse prognosis than EBV-negative DLBCL, even if several studies showed similar outcomes for both forms [90].

Several studies showed that EBV DNA in blood specimens could be a prognostic factor in DLBCL. Liang et al. showed that EBV DNA in whole blood has a good concordance with in situ EBV detection in the tumor and is a better prognostic and monitoring biomarker than EBV tumor status [91]. Tisi et al. investigated whole blood EBV DNA at diagnosis in a cohort of 218 DLBCL patients treated with immunochemotherapy, showing that blood EBV DNA detection is not linked to in situ EBV detection in DLBCL but is instead associated with a worse outcome [92]. Okamoto et al. suggested that EBV DNA detection in pretreatment serum may have an adverse prognostic impact for EBV-positive and EBV-negative DLBCL patients [93]. In a recent large retrospective study with 671 DLBCL patients, the pretreatment presence of EBV DNA in whole blood (>500 copies/mL) had an independent prognostic significance for patients with EBER-positive DLBCL but not for EBER-negative tumors [94]. By contrast, in a study retrospectively analyzing 263 DLBCL patients, high levels of pretreatment EBV DNA in whole blood (>30,800 copies/mL) were associated with a worse prognosis, regardless of tumor EBV status [95].

In conclusion, EBV DNA load may have prognostic significance in DLBCL patients regardless of tumor EBV status, which supports the hypothesis that EBV could disappear from the tumor after contributing to the lymphomagenic process in an earlier phase of cancer development. EBV DNA replication may also be a marker of immune dysfunction, thus contributing to lymphomagenesis by lytic infection from nontumor cells. Given the heterogeneity of studies (i.e., population size, cut-offs for EBER and EBV DNA detection), larger prospective studies are needed to validate the value of EBV DNA as a prognostic factor at diagnosis or as a monitoring biomarker in DLBCL.

Few studies have explored the usefulness of EBV antibody markers in DLBCL, which prevents us from drawing conclusions about a possible link between EBV serological patterns and the risk of DLBCL [96,97,98].

##### Natural Killer/T-Cell Lymphoproliferative Diseases

ENKTL is a rare but highly aggressive type of NHL that is most commonly found in Asian and Latin American populations [72]. This lymphoma is strongly associated with EBV. The association of EBV with the other T-cell lymphoma subtypes is less consistent [71].

In ENKTL patients at diagnosis, EBV DNA is more frequently detected in plasma than in peripheral blood mononuclear cells, thus making plasma a more appropriate specimen for monitoring the therapeutic responses of EBV-NK/T lymphoproliferative diseases [99,100]. In patients with NKTL, plasma EBV DNA is a useful tumor marker for diagnosis, disease monitoring, and prognosis [101]. Pretreatment EBV DNA in plasma and whole blood at diagnosis correlated with lactate dehydrogenase and disease stage. High viral load was associated with poor treatment response and overall survival [94,95,102,103,104,105]. EBV DNA could be a disease marker in patients with ENKTL and kinetics of plasma EBV load were associated with treatment response [100,103]. Plasma EBV DNA detection also appears to be an early indicator of relapse in ENKTL [103,106], whereas undetectable EBV DNA in plasma after chemotherapy is the best predictor of overall survival [104,107].

The humoral immune response against EBV in ENKTL patients has been poorly explored to date. Studies revealed high antibody levels against VCA and EA antigens in patients with ENKTL [108,109]. In contrast with other EBV-associated tumors, anti-EBNA-1 IgG were not elevated in cases of ENKTL [108,109]. A high level of antibodies against EBNA-3A, BZLF-1, and BPLF-1 antigens was detected in ENKTL patients using a multiplex strategy exploring the entire EBV proteome [109]. High anti-EA IgA and anti-VCA IgA levels could thus be related to an adverse ENKTL profile and correlated with poor treatment response, early relapse, and poor prognosis in this population [108].

#### 3.2.2. AIDS-Related Lymphoma

EBV is associated with 20–80% of AIDS-related lymphomas. EBV is detected more frequently in the tumor cells of HIV-infected patients with lymphomas than in HIV-negative patients with the same lymphoma subtype. For example, only 10% of DLBCL, the most common type of NHL, are associated with EBV in the general population, whereas EBV-associated DLBCL represents 40% of all cases in HIV-infected patients. Up to 80–90% of HIV-cHL cases are associated with EBV compared to less than 40% in non-HIV patients [110,111]. Recent studies underlined the synergistic role of HIV and EBV in lymphomagenesis. Beyond HIV-induced immune dysregulation that increases the pool of EBV-infected cells, HIV could interfere indirectly through the induction of chronic B-cell activation or more directly through the interaction of HIV protein P17 with the EBV oncoprotein LMP-1 [112].

Several researchers have examined EBV DNA in peripheral blood as a predictive or prognostic marker in HIV-related lymphomas, although their findings are conflicting due to their different study populations based on combined antiretroviral treatment or HIV-related immune deficiency [113,114,115,116]. In HIV-infected patients with NHL, high levels of pretreatment plasma EBV DNA were a poor prognostic marker, although these results focused only on patients without rituximab treatment [114,117]. By contrast, another study, which only included patients with a recent HIV diagnosis and with optimal management of HIV infection and lymphomas, did not find this association [118]. Two recent studies suggested that EBV DNA load in blood could also be a marker for monitoring patients after chemotherapy in HIV-lymphoma patients [114,118].

A few studies explore the antibody response to EBV in HIV-infected patients with lymphomas, with most showing that patients harbored a high level of IgG antibodies directed against EBV lytic antigens [113,118,119]. In a recent study, high anti-EBNA-1 IgG was associated with a 2.1-fold decreased risk of AIDS-related NHL [119]. In patients in remission 6 months after chemotherapy, a decline in antibody IgG levels against lytic antigens (concomitant to a decrease in EBV DNA load) could point to the restoration of the anti-EBV immune response and the control of EBV replication [113].

The biology of EBV infection could differ in HIV patients with lymphomas compared to non-HIV patients with lymphomas. EBV reactivation and antibody immune response directed against the EBV lytic antigens are frequent in HIV patients with or without lymphoma. This suggests that EBV DNA released from lymphoid or epithelial tissues stems from the EBV lytic cycle and virus replication [41,113,118,120,121]. HIV infection is associated with high levels of chronic B-cell activation, which may affect the EBV load with substantial variations between HIV patients [122]. This variable background of EBV lytic replication is probably associated with EBV DNA derived from lymphoma cells, which may also explain the conflicting results obtained regarding the relevance of EBV DNA load in HIV-lymphoma patients. Indeed, current PCR assays cannot distinguish linear lytic from latent episomal EBV genomes. Molecular techniques capable of differentiating between EBV DNA from tumor cells and EBV virion can be used to determine whether EBV DNA is a prognostic marker in AIDS-related lymphomas, although this process is time-consuming and difficult to implement in routine clinical practice [123].

#### 3.2.3. Post-Transplant Lymphoproliferative Disorders

Most PTLD are associated with EBV: 100% in HSCT recipients and 60–80% in solid organ transplant (SOT) recipients [124]. In HSCT recipients, PTLD mainly derives from donor B lymphocytes, whereas in SOT, they develop from recipient B lymphocytes, except if located in the graft. PTLD are proliferation-related B-cells in 80% of cases. PTLD should be classified using the 2016 WHO histopathological classification (Table 3). The majority are polymorphic and oligoclonal rather than polyclonal.

In EBV-associated PTLD, cell growth is driven by the EBV growth program (also referred to as type III latency) characterized by the expression of (1) nine protein-coding genes (nuclear antigens EBNA-1, -2, -3A, -3B, -3C, LP, and latency membrane proteins LMP-1, -2A, -2B), (2) EBER non-coding RNAs, and (3) miRNAs (Table 1). These latent proteins and RNA as well as EBV lytic cycle proteins like ZEBRA (also known as Zeta or EB-1), may play a role in lymphomagenesis. In transplant recipients, this uncontrolled cell growth is mainly due to the lack of immune surveillance, although genetic background and chronic inflammation probably play a role [3,124].

PTLD diagnosis is essentially based on clinical symptoms, imaging, and anatomopathological analysis. Symptoms may include IM (fever, fatigue, polyadenopathy, tonsillitis, or pharyngitis), organ-specific diseases (depending on the location), graft dysfunction in SOT, and hepatic or hematologic disorders for early PTLD. In later forms of the disease, symptoms are more severe, with polyvisceral involvement mimicking sepsis or severe graft-versus-host disease. Imaging aims to detect tumors or abnormal tissue. Anatomopathological analysis allows for PTLD classification and detection of EBV mRNA expression by in situ hybridization (EBERs) or immunohistochemistry (EBNA or LMP-1 protein) in the tumor cells. EBV viral load in the tumor has no diagnostic value, although it can be useful if the anatomopathological results are difficult to interpret.

EBV viral load in whole blood or plasma is not useful for PTDL diagnosis, although it can help in the monitoring of transplant patients for the early detection and treatment of PTLD or even anticipate treatment needs and thus improve disease prognosis. The monitoring of EBV viral load in whole blood or plasma is driven by PTLD risk factors. For example, EBV-seronegative patients who receive a graft from EBV-seropositive donors, lung or intestine transplant recipients, and recipients with graft rejection or graft-versus-host disease are eligible for viral load monitoring. European and international guidelines recommend weekly follow-up from the first week to the fourth month after transplantation followed by monthly or bimonthly monitoring for 1 year. For HSCT recipients, the risk of PTLD is 0.2% during the first month, so follow-up can begin thereafter. Monitoring can be reinforced or prolonged depending on the evolution of risk factors [125,126].

Despite its usefulness, EBV DNA load has an imperfect predictive value. Indeed, patients with PTDL usually present a high EBV load, although some transplant recipients without PTLD can also have a persistently high EBV load of more than 10,000 copies/mL. Thus, knowledge of EBV DNA kinetics is essential, since a sudden increase (more than 10 copies/mL) or a steady increase over several successive samples can be indicative of PTLD [75]. As mentioned above in relation to HIV-lymphoma patients, EBV DNA may result from EBV-infected B lymphocyte proliferation, which is predictive of both PTLD and virus replication, regardless of the association with lymphoproliferation. Currently, no straightforward technique is available to differentiate the intracellular episomal genome (found in latent infections and lymphoproliferations) from the viral linear genome (present in productive infections). Several studies suggest that the evaluation of EBV genome methylation may be an interesting tool, as low methylation levels point to a productive replication process, whereas high methylation levels indicate a latent/proliferative process [123,127,128]. This will be detailed in the final section below.

Several technical limitations prevent the optimal use of EBV DNA in peripheral blood. First, laboratories use a variety of tools (e.g., extraction and PCR methods or kits, instruments, platforms, targeted genes), which leads to a substantial variability in viral load results. The development of the International Standard for Epstein–Barr Virus in 2014 improved the commutability of results between laboratories [129]. Nevertheless, it is still recommended to monitor EBV load in the same laboratory using the same technique and to interpret the viral load kinetics rather than a single EBV DNA measurement. Second, there is no clear consensus regarding the sample type to be used. EBV load in whole blood appears earlier than in plasma or serum, but it is less specific regarding PTLD risk. Analyzing both sample types (whole blood + plasma or serum) may contribute to the virological diagnosis [130]. Third, there is a lack of consensus regarding therapeutic thresholds, even if researchers recommend 1000 copies/mL in plasma and 5000 copies/mL in whole blood [131,132].

In addition to the early detection of PTLD, the follow-up of EBV load can help clinicians to optimize the immunosuppressive treatment, which likewise contributes to PTLD prevention [41].

Aside from EBV DNA load, other markers are currently being evaluated to help with therapeutic decision-making. Evaluating T-cell immunity against EBV using ELISPOT techniques or IFN-gamma assays can show the patient’s ability to mount a defense against lymphoproliferation. No commercialized tests currently exist, although some studies use a predictive immunological and virological score (CLIV score) to monitor PTLD risk [133]. Although EBV serology is of little value in the early detection of PTLD, the detection of EBV lytic cycle transcripts or proteins such as the ZEBRA protein may be of interest. Several studies point to the predictive prognostic value of ZEBRA detection and its usefulness to investigate EBV lytic infection in transplant patients [134,135].

Finally, host or EBV miRNA (BART, BHRF-1) may be good candidates as markers for PTLD. This will be discussed below [136,137,138].

Regarding PTLD, particular attention should be paid to EBV mucocutaneous ulcers, which were first described as a type of lymphoproliferative disorder [139,140,141,142]. The 2016 WHO classification [143] nevertheless recognized EBV mucocutaneous ulcers as a distinct clinicopathological entity. In the context of transplantation (only 10% of cases), there is a risk of confusing EBV mucocutaneous ulcers with PTLD and even more with HL, since these distinct entities have very similar virological and immunophenotypic markers. EBV mucocutaneous ulcers occur in the absence of a tumor mass; no other sites are involved, and there is no adenopathy. The pathology is characterized by polymorphous infiltrate and atypical large B-cell blasts that co-express B-cell antigens and CD30, often with Hodgkin Reed–Sternberg cell-like morphology [126].

## 4. Immune-Mediated Inflammatory Diseases (IMIDs)

In contrast to EBV-associated malignancies where the role of EBV is well established, the association of the virus with autoimmune diseases, other than MS, is more controversial. The mechanisms by which EBV infection might trigger autoimmune diseases remain elusive, although various hypotheses have been proposed [144].

### 4.1. EBV Infection and Multiple Sclerosis

MS is the most prevalent chronic inflammatory and neurodegenerative disease of the central nervous system characterized by inflammatory demyelination and axonal damage, leading to irreversible neurological disability in young adults. Worldwide, approximately 2.8 million people have MS (prevalence 3–300/100,000 depending on the region) [145]. MS is an autoimmune disease with complex interactions between genetic susceptibility factors (in particular, the HLA-DRB1*15.01 allele), sex and environmental factors (including infectious agents), vitamin D deficiency, childhood obesity, tobacco smoking, and other factors influencing the immune system. Epidemiological and biological evidence currently supports the hypothesis that genetic predispositions, possibly with some environmental factors, alter the host immune response to EBV primary infection and persistence, meaning that the virus is the main trigger of MS onset and possibly one aspect of disease progression [146,147,148].

The most important epidemiological evidence supporting the role of EBV in MS onset was provided by Aschiero et al., who recently gathered convincing data about the increasing risk of MS with EBV seropositivity, EBV seroconversion (see below), high anti-EBNA-1 antibody titers, and IM [149]. In 2022, drawing on the 20-year follow-up of 10 million American army personnel, they demonstrated a 32-fold increased risk of MS in individuals who seroconverted to EBV seropositivity compared to those who remained seronegative. These seroconversions also preceded an increase in serum neurofilament light chain, which is an early biomarker of neurodegeneration in patients who later developed overt MS [149].

Several non-exclusive but plausible mechanisms have been proposed to explain EBV in the pathogenesis of MS. Data from Italy demonstrated the presence of EBV infected cells in post-mortem brain samples from MS patients, mainly in B cells and plasma cells associated with the infiltration of specific anti-EBV cytotoxic T cells [147]. This persistent viral infection in the central nervous system is located preferentially in active MS lesions and may directly or indirectly contribute to oligodendrocyte injury. Interestingly, besides EBV latent proteins, EBV lytic proteins (ZEBRA) are also expressed in plasma B cells, although the role of lytic infection in MS pathogenesis is still unknown. Molecular mimicry in which EBV infection induces antibody or T-cell responses against viral antigens that cross-react with myelin basic protein and other central nervous system antigens (i.e., anoctamin, α-β crystallin) is an important mechanism in MS pathogenesis. Recently, in the cerebrospinal fluid and blood of MS patients, Robinson et al. characterized cross-reacting antibodies against EBNA-1 and the glial cell adhesion molecule, which is a host protein expressed by astrocytes and oligodendrocytes [150]. Moreover, the deleterious neurological effects of this cross-reactivity were previously demonstrated in a mouse model of MS [150].

Numerous other biological mechanisms are currently being explored to decipher the role of EBV infection in MS pathogenesis: the interaction of the viral transcription factor EBNA-2 with the genetic risk loci for MS or vitamin D metabolism; the role of exosomes released by EBV-infected B cells; the consequences of EBV genetic variability; interactions between EBV and other persistent exogenous viral infections (e.g., human herpesvirus 6), and/or endogenous retroviruses [146,147].

The EBV serological assays, DNA viral load measurements, or anti-EBV T-cell activity characterization currently used to monitor EBV infections in the clinical setting are not useful for the detection or follow-up of MS. Nevertheless, analyzing these markers during treatment with highly effective disease-modifying therapies that target B-cell activity during MS may shed light on the role of EBV in MS. Ultimately, the results of ongoing clinical trials on EBV vaccination to prevent EBV infection as well as adoptive T-cell immunotherapy using EBV-specific cytotoxic T lymphocytes to treat MS patients will improve our understanding of the role played by the virus in the initiation and natural history of MS.

### 4.2. Other Autoimmune Diseases

Besides the convincing data about the role of EBV in MS, EBV has long been associated with other autoimmune diseases (AI), specifically systemic lupus erythematosus (SLE), rheumatoid arthritis (RA), and Sjögren syndrome (SS) [151]. As in MS, a higher prevalence of EBV infection, higher titers in serum anti-EBV antibodies, and a higher viral load in blood were reported in SLE, RA, and SS, sometimes correlated with disease activity. The EBV genome or EBV proteins were detected in some but not all synovial fluid specimens from RA patients and salivary gland biopsies from SS patients. Impaired EBV-specific T-cell responses and molecular mimicry between EBV proteins and self-antigens were also documented in SLE and RA. Furthermore, the transcription factor EBNA2 could bind to high-risk alleles of SLE and RA and promote their transcription [152]. Humanized mice with EBV infection were used as an animal model for RA with the induction of erosive arthritis [153]. Currently, serological or molecular EBV assays are not useful for the diagnosis of these diseases or the medical care of patients. Nevertheless, the follow-up of EBV viral load in the blood of AI patients treated with immunosuppressive drugs and at risk of B lymphoproliferative syndromes should be prospectively explored.

## 5. Sequencing, Next-Generation Sequencing, and New EBV Markers

In 1984, the entire double-stranded DNA genome of EBV strain B95-8 and its 172,282 base pairs was sequenced using the time-consuming M13 cloning sequencing technique based on more than 6000 libraries [1]. This paved the way for the discovery of viral latency genes and determined the EBV genome organization and nomenclature. Initially, two distinct EBV types (A and B) with different geographical localizations were identified based on the diversity of latent EBNA genes [154]. Whole genome analysis confirmed this diversity [155], and co-infections with two EBV types were described in patients [156]. In 2013, the Akata and Mutu strains derived from BL were characterized by whole genome sequencing, thus revealing the 171,323 bp and 171,687 bp sequences, respectively, with a few differences compared to B95-8/Raji, GD1, and AG876 strains, notably in terms of the single nucleotide variation in the EBNA latency genes (EBNA-2, EBNA-3A, EBNA-3B, and EBNA-3C). Genomic analysis allowed the identification of RNA editing for miR-BART6 and the non-coding gene BHFL1 [157]. On 29 November 2022, more than 1200 entire or partial EBV genomic sequence data were available in the public National Center for Biotechnology Information database [155,158]. This large number of sequences provides new opportunities for genomic comparison in which their frequent recombination could be the driver of viral evolution [159]. Next-generation sequencing could facilitate the exploration of potentially pathogenic variants in EBV-associated diseases [4,160]. In NPC, whole genome sequencing revealed mutational hotspots in LMP-1 and EBNA-1, thus suggesting that these latent genes are crucial for the pathogenesis of this carcinoma [161].

Recent studies have investigated the use of viral miRNA as potential biomarkers in EBV-associated diseases. EBV encodes 44 mature miRNAs, which are transcribed into two regions around the apoptosis regulator BHRF1 gene (BamHI fragment H rightward reading frame 1) and BART region that controls both viral and cellular genes. On the one hand, EBV-encoded microRNAs help the virus evade immune detection by decreasing the expression of the host immune proteins and immunogenic viral antigens. On the other hand, EBV-encoded microRNAs promote viral replication along with the development of EBV-associated malignancies by epigenetically regulating the expression of molecules that affect apoptosis [162]. miRNAs produced from the EBV gene BART are abundantly expressed in EBV-associated epithelial tumors such as NPC and EBV-associated GC. BART-miRNAs play a critical role in immune escape as well as the development, invasion, and metastasis of NPC. They are divided into two subclusters: subcluster one that includes miR-BART 1, 3–6, and 15–17, and subcluster two that consists of miR-BART 7–14 and 18–21. They are expressed in latency type III infections [162]. A meta-analysis concluded that miRNA expression increased the mortality risk in NPC and GC patients by threefold (hazard ratio: 3.168; 95% CI: 2.020–4.969), meaning they may be used as prognostic biomarkers. Based on the survival data of 711 NPC and 59 GC patients, the meta-analysis investigated a total of seven EBV miRNAs, which were all upregulated [163]. A better understanding of EBV-miRNA expression in EBV-associated GC and NPC will open up new avenues for GC and NPC treatment [163,164]. Circular RNAs are another class of predominantly non-coding RNAs, which may play a role in the viral biology and pathogenicity of EBV-associated malignancies. In the future, serum viral circular RNAs could also be used as potential biomarkers in EBV-associated disorders [165,166].

Finally, epigenetic analysis of EBV DNA has provided new insights into the regulation of viral and cellular gene expression during the viral cycle [167]. EBV DNA methylation patterns varied according to the EBV-associated disease under consideration, which could differ between healthy individuals and patients with NPC [168]. A number of studies analyzing the whole virus methylome or a single gene methylation, showed that EBV methylation was lower in non-malignant tissue biopsies than in malignant tumor biopsies [123,128,169,170]. Lam et al. developed a combined approach of plasma EBV DNA analysis, including the plasma EBV DNA methylome, and the use of the fractional concentration and size of plasma EBV DNA for NPC diagnosis. Borde et al. set up a methyl-qPCR using methyl-sensitive restriction enzymes and PCR on specific regions to differentiate between methylated and unmethylated viral genomes, and to determine the proportion of latent versus lytic viral genome. Whole blood EBV DNA from patients with EBV-associated disease could be more methylated than specimens from healthy individuals [127].

## 6. Conclusions

Although EBV infection is asymptomatic in the majority of individuals, EBV persistence is linked to a wide range of tumors and several immune-mediated inflammatory diseases.

In the absence of vaccines and despite recent advances [171,172,173,174], research efforts should focus on virological markers applicable in clinical practice for the management of patients with EBV-associated diseases. The clinical utility of EBV DNA has been widely studied in EBV-associated malignancies, including NPC in which circulating EBV DNA has been proven to be of clinical value in the screening, prognosis, and surveillance of recurrence for NPC. The usefulness of EBV DNA in lymphoma depends on the type of lymphoma and its association with EBV. Contrary to NPC, few guidelines recommend the use of circulating EBV DNA in the management of lymphoma patients. In HL, NKTL, and DLBCL, several studies showed an association between high blood EBV DNA loads and EBV tumor status or lymphoma prognosis. EBV DNA load also remains the only marker of interest to predict a lymphoproliferative disorder in transplant patients. In AIDS-related lymphomas, given the heterogeneity of studies, the clinical utility of EBV DNA is more controversial. Anti-EBV IgA are still widely used today as biomarkers for the screening of NPC. Atypical serological patterns are described in other EBV-associated malignancies such as EBV-positive HL. However, the clinical utility of EBV serology is limited in pathologies aside from IM and NPC or in the determination of serological EBV status in the context of transplantation. A summary of the clinical utility of EBV markers is shown in Table 4.

New biomarkers are required, particularly in PTLD where EBV DNA lacks specificity. The discrimination between latent EBV DNA from cancer cells and lytic EBV DNA unrelated to the tumoral process may be investigated using technologies exploring the viral methylome. Next-generation sequencing also provides the opportunity to explore the diversity of viral strains or the expression of viral miRNA linked to the higher pathogenicity of EBV-associated diseases. New technologies offer a promising potential for the discovery of other biomarkers and their evaluation in EBV-associated cancers or immune-mediated inflammatory pathologies triggered by EBV.

## Figures and Tables

**Figure 1 viruses-15-00656-f001:**
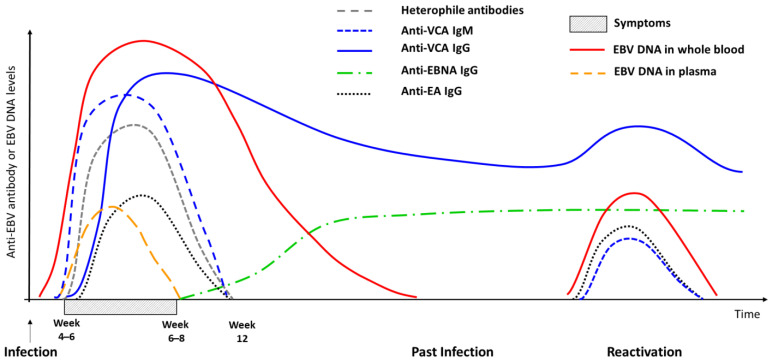
Evolution of antibody and blood EBV DNA markers during EBV infection in an immunocompetent host.

**Table 1 viruses-15-00656-t001:** Main Epstein–Barr virus (EBV)-associated diseases.

EBV-Associated Diseases	EBV Association	Latency Type
Infectious mononucleosis	100%	Latency type III
Carcinomas		
Nasopharyngeal carcinoma	100%	Latency type II
Gastric carcinoma	10%	Latency type I
Lymphomas		
Hodgkin lymphoma *In HIV-infected patients*	30% 90%	Latency type II
Endemic Burkitt lymphoma	>95%	Latency type I
Sporadic Burkitt lymphoma *In HIV-infected patients*	20% 40%	Latency type I
Diffuse large B cell lymphoma	5–15%	Latency type III
Extranodal natural killer/T cell lymphoma	100%	Latency type II
Post-transplant lymphoproliferative disorders SOT HSCT	60–80% 100%	Latency type III
Multiple sclerosis	NA	NA

Latency type I: EBNA1, EBER, miRNA; latency type II: EBNA1, LMP1, LMP2a, 2b, EBER, miRNA; latency type III: EBNA1, 2, 3A, 3B, 3C, LP, LMP1, LMP2a, 2b, EBER, miRNA. Abbreviations: NA: non-applicable; EBNA: Epstein–Barr nuclear antigen; LMP: latent membrane protein; EBER: Epstein–Barr-encoded small RNA; miRNA: micro non-coding RNA encoded by BART and BHRF1 genes; HIV: Human Immunodeficiency Virus; SOT: solid organ transplantation; HSCT: hematopoietic stem cell transplantation; NA: not applicable.

**Table 3 viruses-15-00656-t003:** Histopathological classification of post-transplant lymphoproliferative disorders (PTLD) (2016, WHO classification) and clonal characteristics.

Category		Clonality
Non-Destructive PTLD		Nonclonal
	Plasmacytic hyperplasia	
	Infectious mononucleosis	
	Florid follicular hyperplasia	
Polymorphic PTLD		clonal
Monomorphic PTLD (classified according to the lymphoma to which they correspond)		clonal
	*B-cell neoplasm*	
	Diffuse large B-cell lymphoma	
	Burkitt lymphoma	
	Plasma cell myeloma	
	Plasmacytoma	
	Others *	
	*T-cell neoplasm*	
	Peripheral T-cell lymphoma, not otherwise specified	
	Hepatosplenic T-cell lymphoma	
	Others	
Classic Hodgkin lymphoma PTLD		clonal

* Indolent small B-cell lymphomas arising in transplant recipients are not included as PTLD, with the exception of EBV-positive marginal zone lymphomas.

**Table 4 viruses-15-00656-t004:** Clinical utility of virological markers in EBV-associated diseases.

EBV-Associated Diseases	Anti-EBV Antibodies	Blood EBV DNA
Infectious mononucleosis	Diagnosis	Positive in serum for 15 days
Nasopharyngeal carcinoma	Screening	++ plasma/serum (screening, prognosis, treatment response)
EBV-positive Gastric carcinoma	No	require further evaluations
EBV-positive Hodgkin lymphoma	? (atypical profile)	+ (prognosis)
Endemic Burkitt lymphoma	Research	require further evaluations
Diffuse large B-cell lymphoma	No	+ (prognosis)
Extranodal NK/T cell lymphoma	Research	++ (prognosis, treatment response)
AIDS-related lymphoma	?	require further evaluations (prognosis?)
Post-transplant lymphoproliferative disorders	Serological status	++ (prevention, diagnosis)

Abbreviations: +: clinical utility demonstrated in clinical studies; ++: marker commonly used in clinical practice.

## Data Availability

No new data were created or analyzed in this study. Data sharing is not applicable to this article.

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
