# Peer review of "Virological Markers in Epstein–Barr Virus-Associated Diseases"

_viruses, 2023, doi:10.3390/v15030656_

Round 1

Reviewer 1 Report

In this manuscript, Lupo et al. review molecular markers useful for detecting EBV, how they are being used in the clinic, and future perspectives of the field with emphasis on DNA-based approaches. I find the review interesting and informative, but I have the following questions and suggestions that I would like the authors to consider before publications.

General points:

1)    I understand that the involvement of EBV in proliferative diseases is relatively well-established. In the case of autoimmune diseases, how solid is the evidence for the causal role of EBV? Does it satisfy the Koch’s postulate or Hill’s criteria? I would suggest to state the present status of our understanding more clearly and critically.

2)    Along the same line, it is a little hard for readers to understand, how EBV contributes to tumorigenesis on the one hand and to autoimmune diseases on the other. It would be helpful if the authors include diagrams illustrating the (possible) mechanisms of EBV’s actions in inducing representative diseases (or at least, draw attention to literature presenting such diagrams).

3)    I am not comfortable reading phrases like “Several studies investigating/examined” (lines 76, 366, 428), “Recent studies --- measured” (line 265), “studies report” (lines 345, 365, 403), “studies explored” (line 388, 438), “Studies have found” (409), “studies recommend” (524), “Recent data reported” (line 310), etc. In these phrases, it is not “studies” or “data” that do something (e.g., investigate, examine, report, recommend, etc.): it is RESEARCHERS! Phrases like “studies have shown” (line 115), “Some studies revealed” (line 223) make sense. 

Specific or minor points:

4)    Line 33, “also known”: I guess “as” is missing after this phrase.

5)    Line 35, “almost”: I would suggest to replace this word with a phrase like “the genome of approximately”.

6)    Table 1: 

i.               Abbreviations “HIV” and “NA” should be defined in the legend.

ii.              Usage/meaning of “()” is not consistent in the second column. 

iii.             As for the layout, wider spaces should be left between the end of Table legends and main texts (e.g., between lines 53 and 54). [The same applies to all Tables.]

7)    Line 63, “Some infected B cells --- mechanisms.”: This sentence looks largely redundant to the following sentence. Is it necessary?

8)    Line 70: “Standard markers --- antibodies”: This sentence sounds slightly awkward since the subject of the sentence is “markers”, but it actually describes procedures.

9)     Line 114, “The higher frequency --- HL.” This sentence sound redundant to the following sentence. Is it necessary?

10)  Line 142, “for longer”: I would suggest to insert “period” after this.

11)  Line 181, “HIV primary brain lymphoma”: The “HIV-associated primary central nervous system lymphoma (PCNSL)” is frequently used in the literature. Mentioning this alternative name should be helpful for readers.

12)  Line 200, “made”: I guess “to” is missing after this.

13)  Line 208, “11%”: I am curious whether this number is satisfactory (high enough to be useful) or not. Any comments?

14)  Line 228, “After initiating NPC treatment, --- tumors”: For laymen, “radiosensitive tumors” in the last sentence of this paragraph sounds abrupt. Is radiation commonly used for treating NPC? If so, please mention this earlier (e.g., in the first sentence in this paragraph). 

15)  Line 249, “at baseline”: It is unclear what “baseline” means in this context. Different from “the level”? 

16)  Line 290, “statistically powered”: It is unclear what this actually means.

17)  Lines 296, 416, 459: Are these subheads? If so, I would suggest to number them (e.g., “3.2.1”, etc.).

18)  Line 308, “EBV exists predominantly”: This sounds like almost all EBV-positive cells undergo apoptosis. Is it true?

19)  Line 312, “could be a surrogate marker”: This sounds like one marker could be a surrogate marker for another. Is it of any use?

20)  Line 316, “but not at base-line”: It is unclear what this means. (see my comment 14)

21)  Line 360, “DBLCL” (two times): Are these “DLBCL”?  

22)  Line 404, 519, “kinetic”: Is this “kinetics”?

23)  Line 411, “A high rate”: It is unclear what the “rate” means in this context.

24)  Line 501, “above than 4 log”: This should be either “above” or “more than”. Also, “104” might be easier to understand than “4 log”. (also see line 502, “1 log”)

25)  Line 520, “matrix” and line 522, “matrices”: I am not familiar with this usage of the words. Are they commonly used in this field? How about using more common word like “material” or “sample”? Not quite?

26)  Line 523, “and in connection with first and second statements”: It is hard for readers to find these statements. It might be sufficient to say “and in connection with above statements”, or I would just delete the whole insert.

27)  Line 566, “seroconversion”: I would insert “(see below)” after this word, since at first sight, it is unclear what this technical term means, although it became clear as we read the following sentences. 

28)  Line 611, “Furthermore, --- reported.”: It is hard to understand what is written in this sentence, although it sounds interesting. At least, citation of the original paper after this sentence should be helpful.

29)  Line 633, “genomes were available”: This sounds awkward. What were available should be “genomic sequence data” not “genomes” themselves.

30)  Line 639, “pathogenicity of this carcinoma”: This also sounds awkward. It should be either “pathogencity of this virus” or “pathogenesis of this carcinoma”.

31)  Line 654, “by threefold”: It is hard to evaluate whether this number is meaningful. It would be better to provide statistical values such as N and P.

32)  Line 672, “to differentiate the quantification of”: This sounds a litter strange. I would put it like “to differentiate between, and to determine the proportion of,”.

Author Response

Response to the Reviewers

Manuscript ID: viruses-2203614

Type of manuscript: Review

Title: Virological markers in Epstein-Barr virus-associated diseases

Authors : Julien LUPO*, Aurélie Truffot, Julien Andreani, Mohammed Habib, Olivier Epaulard, Patrice Morand, and Raphaële Germi

We thank the Reviewers for their positive evaluation of our manuscript. We are grateful for their comments and suggestions, which allowed us to improve the quality of the manuscript. The Reviewers’ comments were carefully addressed, as detailed below. Additions and changes made in the text can be viewed in the co-submitted marked version of the revised manuscript (named « Manuscript_marked »). Please note that the page and line numbers listed below refer to the marked version of the revised manuscript.

Reviewers' comments:

Reviewer 1

Comments and Suggestions for Authors

In this manuscript, Lupo et al. review molecular markers useful for detecting EBV, how they are being used in the clinic, and future perspectives of the field with emphasis on DNA-based approaches. I find the review interesting and informative, but I have the following questions and suggestions that I would like the authors to consider before publications.

General points:

  • I understand that the involvement of EBV in proliferative diseases is relatively well-established. In the case of autoimmune diseases, how solid is the evidence for the causal role of EBV? Does it satisfy the Koch’s postulate or Hill’s criteria? I would suggest to state the present status of our understanding more clearly and critically.
  • Along the same line, it is a little hard for readers to understand, how EBV contributes to tumorigenesis on the one hand and to autoimmune diseases on the other. It would be helpful if the authors include diagrams illustrating the (possible) mechanisms of EBV’s actions in inducing representative diseases (or at least, draw attention to literature presenting such diagrams).

Response

We thank the reviewer for its constructive comments. Our paper is not a comprehensive view of the pathophysiology of EBV-associated diseases. Its main aim is to focus on the clinical utility of EBV markers. Only a brief basic introduction to the pathology precedes each section. We agree that the role of EBV in tumorigenesis has been well established for many years, whereas its implication in autoimmune diseases is more controversial. Recent studies have provided convinced evidences of an association between EBV and MS. For the other autoimmune disorders, further studies are needed to strengthen the link with EBV. The mechanisms involved in EBV-associated tumorigenesis and autoimmune disorders are different and should be discussed.

Accordingly, we added some sentences in the Introduction and in Section 4.2 (other autoimmune diseases). See lines 71-73 and 568-573. We added two citations. Mesri et al. (Cell host Microbes 2014; Human Viral Oncogenesis: A Cancer Hallmarks Analysis) proposed an excellent diagram explaining the mechanisms of oncogenesis in EBV-associated-diseases. Regarding the role of EBV in autoimmune diseases, a recent review by Zhang (J Med Virol 2023) provides a schematic explanation of the putative mechanisms linking EBV infection to autoimmune diseases.

  • I am not comfortable reading phrases like “Several studies investigating/examined” (lines 76, 366, 428), “Recent studies --- measured” (line 265), “studies report” (lines 345, 365, 403), “studies explored” (line 388, 438), “Studies have found” (409), “studies recommend” (524), “Recent data reported” (line 310), etc. In these phrases, it is not “studies” or “data” that do something (e.g., investigate, examine, report, recommend, etc.): it is RESEARCHERS! Phrases like “studies have shown” (line 115), “Some studies revealed” (line 223) make sense. 

Response:

We thank the reviewer for its suggestions. We modified these sentences as requested.

Specific or minor points:

4)    Line 33, “also known”: I guess “as” is missing after this phrase.

5)    Line 35, “almost”: I would suggest to replace this word with a phrase like “the genome of approximately”.

6)    Table 1: 

  1. Abbreviations “HIV” and “NA” should be defined in the legend.
  2. Usage/meaning of “()” is not consistent in the second column. 

iii.             As for the layout, wider spaces should be left between the end of Table legends and main texts (e.g., between lines 53 and 54). [The same applies to all Tables.]

7)    Line 63, “Some infected B cells --- mechanisms.”: This sentence looks largely redundant to the following sentence. Is it necessary?

8)    Line 70: “Standard markers --- antibodies”: This sentence sounds slightly awkward since the subject of the sentence is “markers”, but it actually describes procedures.

9)     Line 114, “The higher frequency --- HL.” This sentence sound redundant to the following sentence. Is it necessary?

10)  Line 142, “for longer”: I would suggest to insert “period” after this.

11)  Line 181, “HIV primary brain lymphoma”: The “HIV-associated primary central nervous system lymphoma (PCNSL)” is frequently used in the literature. Mentioning this alternative name should be helpful for readers.

12)  Line 200, “made”: I guess “to” is missing after this.

13)  Line 208, “11%”: I am curious whether this number is satisfactory (high enough to be useful) or not. Any comments?

14)  Line 228, “After initiating NPC treatment, --- tumors”: For laymen, “radiosensitive tumors” in the last sentence of this paragraph sounds abrupt. Is radiation commonly used for treating NPC? If so, please mention this earlier (e.g., in the first sentence in this paragraph). 

15)  Line 249, “at baseline”: It is unclear what “baseline” means in this context. Different from “the level”? 

16)  Line 290, “statistically powered”: It is unclear what this actually means.

17)  Lines 296, 416, 459: Are these subheads? If so, I would suggest to number them (e.g., “3.2.1”, etc.).

18)  Line 308, “EBV exists predominantly”: This sounds like almost all EBV-positive cells undergo apoptosis. Is it true?

19)  Line 312, “could be a surrogate marker”: This sounds like one marker could be a surrogate marker for another. Is it of any use?

20)  Line 316, “but not at base-line”: It is unclear what this means. (see my comment 14)

21)  Line 360, “DBLCL” (two times): Are these “DLBCL”?  

22)  Line 404, 519, “kinetic”: Is this “kinetics”?

23)  Line 411, “A high rate”: It is unclear what the “rate” means in this context.

24)  Line 501, “above than 4 log”: This should be either “above” or “more than”. Also, “104” might be easier to understand than “4 log”. (also see line 502, “1 log”)

25)  Line 520, “matrix” and line 522, “matrices”: I am not familiar with this usage of the words. Are they commonly used in this field? How about using more common word like “material” or “sample”? Not quite?

26)  Line 523, “and in connection with first and second statements”: It is hard for readers to find these statements. It might be sufficient to say “and in connection with above statements”, or I would just delete the whole insert.

27)  Line 566, “seroconversion”: I would insert “(see below)” after this word, since at first sight, it is unclear what this technical term means, although it became clear as we read the following sentences. 

28)  Line 611, “Furthermore, --- reported.”: It is hard to understand what is written in this sentence, although it sounds interesting. At least, citation of the original paper after this sentence should be helpful.

29)  Line 633, “genomes were available”: This sounds awkward. What were available should be “genomic sequence data” not “genomes” themselves.

30)  Line 639, “pathogenicity of this carcinoma”: This also sounds awkward. It should be either “pathogencity of this virus” or “pathogenesis of this carcinoma”.

31)  Line 654, “by threefold”: It is hard to evaluate whether this number is meaningful. It would be better to provide statistical values such as N and P.

32)  Line 672, “to differentiate the quantification of”: This sounds a litter strange. I would put it like “to differentiate between, and to determine the proportion of,”.

Response:

We thank the reviewer for its careful proofreading. We corrected the errors throughout the text and we modified all confusing sentences as suggested by the reviewer. Subheadings were numbered. Wider spaces were added between the table legends and the main text.

Comments requiring specific responses:

#6. Table 1: 

  1. Abbreviations “HIV” and “NA” should be defined in the legend.
  2. Usage/meaning of “()” is not consistent in the second column. 

iii.  As for the layout, wider spaces should be left between the end of Table legends and main texts (e.g., between lines 53 and 54). [The same applies to all Tables.]

Response: We modified the Table 1 as suggested. Two new rows were specifically created for HIV patients.

#13. Line 208, “11%”: I am curious whether this number is satisfactory (high enough to be useful) or not. Any comments?

Response: The positive predictive value of 11% is correct, although it seems low. This value depends on the prevalence of the disease in the population. If the prevalence is low, (which is the case for cancer screening studies in asymptomatic populations), the positive predictive value will be low but the negative predictive value will be high. Contrary to “our intuition”, this value is relatively good and superior to other studies that performed cancer screening in asymptomatic populations. We added the negative predictive value found in this study for information. Of the 20,174 participants in the study only 309 had detectable plasma EBV DNA at baseline and at follow-up and of these 309 participants, NPC was confirmed in 34 (11%).

#14. Line 228, “After initiating NPC treatment, --- tumors”: For laymen, “radiosensitive tumors” in the last sentence of this paragraph sounds abrupt. Is radiation commonly used for treating NPC? If so, please mention this earlier (e.g., in the first sentence in this paragraph).

Response: As suggested, we added a sentence on NPC treatment, which bases on radiotherapy.

#15. Line 249, “at baseline”: It is unclear what “baseline” means in this context. Different from “the level”?

Response: To avoid confusion, we changed the term “at baseline” to “initially”.

#16. Line 290, “statistically powered”: It is unclear what this actually means.

Response: We modified as follows: “larger studies with adequate statistical power”

#18. Line 308, “EBV exists predominantly”: This sounds like almost all EBV-positive cells undergo apoptosis. Is it true?

Response: We changed the sentence: “EBV exists in serum or plasma as episomal or naked EBV DNA derived from predominantly lymphoma cells during the process of apoptosis/necrosis”

#19. Line 312, “could be a surrogate marker”: This sounds like one marker could be a surrogate marker for another. Is it of any use?

Response: EBV DNA could be a surrogate marker; its analytical performance is excellent in terms of reproducibility, sensitivity and specificity (it depends on the cut off for EBV DNA detection) and it is easier to implement than EBER detection. However, in routine clinical practice, EBER detection (in situ EBV detection) remains the gold standard for determining that a cancer is associated with EBV. To avoid confusion we modified the terms “could be a surrogate marker” to “correlated with”.

#20. Line 316, “but not at base-line”: It is unclear what this means. (see my comment 14)

Response: We changed “at baseline” for ‘at diagnosis”

#23. Line 411, “A high rate”: It is unclear what the “rate” means in this context.

Response: We changed the word for “level”

#25. Line 520, “matrix” and line 522, “matrices”: I am not familiar with this usage of the words. Are they commonly used in this field? How about using more common word like “material” or “sample”? Not quite?

Response: We modified for “sample type”

#28. Line 611, “Furthermore, --- reported.”: It is hard to understand what is written in this sentence, although it sounds interesting. At least, citation of the original paper after this sentence should be helpful.

Response: We changed the sentence to make it more comprehensible and added the reference of the original paper (Harley and al., Nat Genetics 2018).

# 31. Line 654, “by threefold”: It is hard to evaluate whether this number is meaningful. It would be better to provide statistical values such as N and P.

Response: We added the statistical values, as requested (hazard ratio and 95% CI)

Reviewer 2 Report

This is a very good and thorough review of biomarkers for detection of EBV in diseases associated with this ubiqitous virus. It mainly focuses on detection of EBV in various cancers, but multiple sclerosis is also included. The role of EBV in systemic rheumatic autoimmune diseases is only briefly mentioned, which is the only weakness of the paper.

Minor points.

Lines:

40 encode

200 made to identify

200 or in an early

258 gp350 envelope can (delete "glycoprotein" this is already in "gp350" (maybe explain abbreviation))

267 anti-EBNA IgA (delete "antibodies", this is already in "anti-" and in "IgA", no need for 3 x antibodies)

283 causally ? (not "casually" !)

397 cells

404 kinetics

411 delete "antibodies" (same as in line 267)

454 current PCR 

463 proliferation-related

501 above 4 log

523 with the first

531 Quantiferon test (maybe better with "IFN-gamma assays"

531 ability to mount a defence

563 et al.,

684 EBV DNA has been proven

693 delete "antibodies"

573 Data from Italy demonstrated 

Tables: Insert space after end of table to make the text stand out more clearly

Author Response

Response to the Reviewers

Manuscript ID: viruses-2203614

Type of manuscript: Review

Title: Virological markers in Epstein-Barr virus-associated diseases

Authors : Julien LUPO*, Aurélie Truffot, Julien Andreani, Mohammed Habib, Olivier Epaulard, Patrice Morand, and Raphaële Germi

We thank the Reviewers for their positive evaluation of our manuscript. We are grateful for their comments and suggestions, which allowed us to improve the quality of the manuscript. The Reviewers’ comments were carefully addressed, as detailed below. Additions and changes made in the text can be viewed in the co-submitted marked version of the revised manuscript (named « Manuscript_marked »). Please note that the page and line numbers listed below refer to the marked version of the revised manuscript.

Reviewer 2

Comments and Suggestions for Authors

This is a very good and thorough review of biomarkers for detection of EBV in diseases associated with this ubiqitous virus. It mainly focuses on detection of EBV in various cancers, but multiple sclerosis is also included. The role of EBV in systemic rheumatic autoimmune diseases is only briefly mentioned, which is the only weakness of the paper.

Response:

We thank the reviewer for its positive assessment. As our paper focuses on EBV markers and not on the pathophysiology of EBV-associated diseases, we did not address the role of EBV in systemic rheumatic autoimmune diseases although it is of interest. We added an interesting reference on the role of the EBV transcription factor EBNA-2 in SLE, which provides explanations on the underlying molecular mechanisms between EBV and autoimmune diseases (Harley et al. Nat Genetics 2018) and another reference on the role of EBV in rheumatoid arthritis (Kuwana et al., Plos One 2011).

Minor points.

Lines:

40 encode

200 made to identify

200 or in an early

258 gp350 envelope can (delete "glycoprotein" this is already in "gp350" (maybe explain abbreviation))

267 anti-EBNA IgA (delete "antibodies", this is already in "anti-" and in "IgA", no need for 3 x antibodies)

283 causally ? (not "casually" !)

397 cells

404 kinetics

411 delete "antibodies" (same as in line 267)

454 current PCR 

463 proliferation-related

501 above 4 log

523 with the first

531 Quantiferon test (maybe better with "IFN-gamma assays"

531 ability to mount a defence

563 et al.,

684 EBV DNA has been proven

693 delete "antibodies"

573 Data from Italy demonstrated 

Tables: Insert space after end of table to make the text stand out more clearly

Responses to minor points:

We are grateful to the reviewer for its careful proofreading; we followed all its recommendations. The errors were corrected throughout the text and spaces were inserted at the end of tables and legends.

Reviewer 3 Report

This is Review article about amrkers in EBV-associated diseases.   I believe that the authors have described infectious mononucleosis too extensively. This section should be shortened. More attention should be paid to EBV-associated cancers.

The Authors wrote: “EBV is associated with undifferentiated NPC, which is endemic to Southern China 190 and Southeast Asia”.

That's true. However, European researchers have also found EBV in nasopharyngeal cancer. Various biomarkers has been tested in EBV-associated oropharyngeal cancer. 

e.g. manuscripts  

1.Salivary and serum IL-10, TNF-α, TGF-β, VEGF levels in oropharyngeal squamous cell carcinoma and correlation with HPV and EBV infections. Infect. Agents Cancer 2016, 11, 45

2.Serum and Tissue Level of TLR9 in EBV-Associated Oropharyngeal Cancer Cancers 2021, 13(16), 3981;

3.Cytokines in saliva as biomarkers of oral and systemic oncological or infectious diseases: A systematic review,Cytokine,Volume 143,2021,155506,   In most studies, the authors detected EBV DNA in paraffin-embedded tissue. Others in fresh frozen tissue. The frequency of detection depends on the type of material.   The paragraph “Gastric Carcinoma”has been described very fragmentarily     These are just some articles. I believe that studies on different biomarkers that have been conducted in different centers should be presented. Therefore, I believe that the manuscript should be supplemented and redrafted.  

Author Response

Response to the Reviewers

Manuscript ID: viruses-2203614

Type of manuscript: Review

Title: Virological markers in Epstein-Barr virus-associated diseases

Authors : Julien LUPO*, Aurélie Truffot, Julien Andreani, Mohammed Habib, Olivier Epaulard, Patrice Morand, and Raphaële Germi

We thank the Reviewers for their positive evaluation of our manuscript. We are grateful for their comments and suggestions, which allowed us to improve the quality of the manuscript. The Reviewers’ comments were carefully addressed, as detailed below. Additions and changes made in the text can be viewed in the co-submitted marked version of the revised manuscript (named « Manuscript_marked »). Please note that the page and line numbers listed below refer to the marked version of the revised manuscript.

Reviewer 3

This is Review article about amrkers in EBV-associated diseases.   I believe that the authors have described infectious mononucleosis too extensively. This section should be shortened. More attention should be paid to EBV-associated cancers.

Response:

We thank the reviewer for its constructive criticism and its evaluation. As suggested, we shortened the section dealing with IM by removing the sentences describing the complications of IM (i.e. about 75 words).

The Authors wrote: “EBV is associated with undifferentiated NPC, which is endemic to Southern China 190 and Southeast Asia”.

That's true. However, European researchers have also found EBV in nasopharyngeal cancer.

Various biomarkers has been tested in EBV-associated oropharyngeal cancer. 

e.g. manuscripts  

1.Salivary and serum IL-10, TNF-α, TGF-β, VEGF levels in oropharyngeal squamous cell carcinoma and correlation with HPV and EBV infections. Infect. Agents Cancer 2016, 11, 45

2.Serum and Tissue Level of TLR9 in EBV-Associated Oropharyngeal Cancer Cancers 2021, 13(16), 3981;

3.Cytokines in saliva as biomarkers of oral and systemic oncological or infectious diseases: A systematic review,Cytokine,Volume 143,2021,155506

   In most studies, the authors detected EBV DNA in paraffin-embedded tissue. Others in fresh frozen tissue. The frequency of detection depends on the type of material.

Response:

We thank the reviewer for opening our mind to other putative EBV-associated carcinomas such as oropharyngeal cancer. To our knowledge, it is well established that EBV is a key factor in the development of NPC whereas the association between EBV and oropharyngeal cancer is less well documented (in contrast to HPV or other factors such as tobacco, alcohol). However, as the nasopharynx and oropharynx are anatomically and structurally adjacent areas of the pharynx, it is a good idea to investigate the possible role of EBV in oropharynx cancer and such work will be encouraged. Note that we have voluntarily not included breast cancer, where the association with EBV remains controversial.

We also would like to remind you that our paper focuses on virological markers that can be used in clinical practice to manage patients. The question of non-virological markers, although of interest, (e.g. cytokines) is deliberately not addressed in this review. We added a sentence in the manuscript (NPC section) to mention that EBV could be an environmental factor in the development of oropharyngeal carcinoma and we added the reference Stepien et al., Cancers 2021.

   The paragraph “Gastric Carcinoma”has been described very fragmentarily     These are just some articles. I believe that studies on different biomarkers that have been conducted in different centers should be presented. Therefore, I believe that the manuscript should be supplemented and redrafted.  

Response:

According to our literature search, significant contributions dealing with virological markers useful for the management of patients with EBV-associated GC (this is the scope of our paper) are not common. New markers (EBV-miRNA) in GC are already discussed in section 5. We found one additional reference that is consistent with the scope of the paper (we cited Shoda et al., Oncotarget 2017).